# The First Russian Patient with Native American Myopathy

**DOI:** 10.3390/genes13020341

**Published:** 2022-02-13

**Authors:** Aysylu Murtazina, Nina Demina, Polina Chausova, Olga Shchagina, Artem Borovikov, Elena Dadali

**Affiliations:** Research Centre for Medical Genetics, 115478 Moscow, Russia; demina@med-gen.ru (N.D.); polinaalex85@gmail.com (P.C.); schagina@dnalab.ru (O.S.); borovikov33@gmail.com (A.B.); genclinic@yandex.ru (E.D.)

**Keywords:** congenital myopathy, Native American myopathy, STAC3, NAM

## Abstract

Congenital myopathy associated with pathogenic variants in the *STAC3* gene has long been considered native American myopathy (NAM). In 2017, the first case of a non-Amerindian patient with this myopathy was described. Here, we report the first Russian patient with NAM. The patient is a 17-year-old female with compound-heterozygous single nucleotide variants in the *STAC3* gene: c.862A>T, p.(Lys288Ter) and c.93del, p.(Lys32ArgfsTer78). She has a milder phenotype than the earlier described patients. To our knowledge, this is the first case of a patient who had both nonsense and frameshift variants. It is assumed that the frameshift variant with premature stop codon lead to nonsense-mediated RNA decay. However, there are two additional coding isoforms of the *STAC3* gene, which are not affected by this frameshift variant. We can speculate that these isoforms may partially carry out the function, and possibly explain the milder phenotype of our patient.

## 1. Introduction

Congenital myopathies (CM) are a clinically and genetically heterogenic group of muscle disorders, united by the early onset of muscle hypotonia and muscle weakness at birth or infancy. The group of CM constantly expands due to the revealing of new associations with new genes. A relatively new gene *STAC3* is associated with Native American myopathy (NAM), in which until 2017, is described only in patients from the Lumber River of North Carolina [1,2]. To date, despite the name of the disease, more than 20 non-Amerindian cases of this myopathy are published [3,4,5].

The first description of NAM was attributed to Bailey and Bloch, who in 1987 described this myopathy in a 3-month-old Indian baby with malignant hyperthermia [6]. Malignant hyperthermia is one of the major features of congenital myopathy caused by variants in the *STAC3* gene [5]. In addition to muscular manifestations, patients with NAM have a specific craniofacial phenotype, short stature, scoliosis, talipes equinovarus, often presenting with hearing loss and dental anomalies, as well as cryptorchidism in males [1,5].

Here, we present a non-Amerindian patient of Russian ancestry, with compound heterozygous single nucleotide variants (SNVs) in the *STAC3* gene (ENST00000332782.7): c.862A>T, p.(Lys288Ter) and c.93del, p.(Lys32ArgfsTer78). Our patient demonstrated early developed muscle weakness, some facial features, and severe scoliosis, but a lack of peculiar features, such as short stature, palate anomalies, multiple joint contractures, and feeding difficulties in infancy, exhibiting a milder phenotype than in the earlier reported cases.

## 2. Case Report

The proband is a 17-year-old female with the chief complaint of proximal muscle weakness from early childhood. She is the second child of healthy nonconsanguineous Russian parents. The pregnancy and delivery were unremarkable. Her birth weight was 3.80 kg and length was 52 cm. On the third day of life, the girl was hospitalized with pneumonia due to aspiration. She required mechanical ventilation until the age of 4.5 months, and it was the only episode of respiratory disfunction in her life. The proband had motor development delay, she held her head from 7 months of age, sat from 16 months of age, walked from 21 months of age, and she always required support to stand up. In early childhood, it was noted that her mouth was slightly open in a vertical position. Phrasal speech developed by age. There was no history of any seizures. Vision and hearing were preserved. From the age of 9–10 years, the development of prominent scoliotic deformity of the spine was noted. A conservative treatment of scoliosis was carried out at the age of 12–14 years. At the age of 12, she underwent surgery for ankle tendon retraction without general anesthesia. In the last few years, the course of her disease has become slowly worse. In addition, she had additional marked gait disturbances and difficulties climbing the stairs, and she could not rise from the floor by herself. Her cognitive functions were normal, and currently she is studying at university.

At the examination, her height was 167 cm and body weight was 62 kg. Among the phenotypic features, the proband has only micrognathia, with downturned corners of the mouth and clinodactyly of the fifth finger of the right hand (Figure 1).

A neurological exam showed the atrophy of the pectoral muscles, proximal muscles, weakness of facial, neck, and limb-girdle muscles, positive Gowers’ sign, and asymmetric Trendelenburg’s gait. The patient showed kyphoscoliosis of the thoracic spine and lumbar hyperlordosis that led to a prominent asymmetric hip position (Figure 1). Moreover, she had pes cavus on both sides. The leg tendon reflexes were absent, and the biceps and triceps reflexes were normal.

Creatine kinase values were normal. A needle electromyography detected a myopathic pattern and a spontaneous activity was absent.

For diagnostic purposes, we performed a next-generation sequencing of the genes panel on DNA, which is extracted from the proband’s blood lymphocytes. The sample was analyzed by mass parallel sequencing using a new-generation Ion S5™ sequencer (Thermo Fisher Scientific Inc., Waltham, MA, USA). The libraries for sequencing were prepared with ultramultiplex PCR (AmpliSeq™). The target panel based on Ion Ampliseq technology included the following genes: *SELENON, LMNA, TPM3, ACTA1, PLOD1, MYPN, ITGA7, STAC3, CNTN1, MYF6, CFL2, KBTBD13, TRIP4, CHST14, CCDC78, MYH2, FKRP, TNNT1, RYR1, DNM2, COL6A3, KLHL41, BIN1, ZAK, SPEG, COL5A2, COL6A1, COL6A2, CHKB, ITGA9, KLHL40, LMOD3, MTMR14, MEGF10, LAMA2, COL12A1, DSE, FKBP14, AEBP1, TPM2, FKTN, COL5A1, MTM1,* and *VMA21*. The coverage of target variants for DNA was at least ×200. The detected variants were analyzed using an hg19 genome assembly, HGMD Professional Database v2020.2, and ACMG criteria for variant interpretation [7]. The detected variants were validated by Sanger sequencing. The sequencing was carried out using the ABI Dye Terminator, v3.1. (Applied Biosystems), on a 3130 × l ABI genetic analyzer (Applied Biosystems). The reference cDNA sequence was taken from the GenBank database. The analysis revealed two heterozygous SNVs: c.862A>T, p.(Lys288Ter) and c.93del, p.(Lys32ArgfsTer78) in the *STAC3* gene (ENST00000332782.7). The validation and segregation analyses of detected SNVs showed that each of the parents is a carrier of only one, which verifies the compound heterozygosity of SNVs in the proband. The healthy brother is a carrier of the heterozygous variant c.862A>T, p.(Lys288Ter).

## 3. Discussion

Herein, we reported a patient of Russian ancestry diagnosed with a rare form of congenital myopathy. Notably, our patient does not have a severe clinical picture and lacks some characteristic phenotypic features, as previously described in the literature (Table 1).

Patients with NAM present quite specific extra-muscular manifestations along with the common signs of congenital myopathy, such as the long myopathic face, palate and dental anomalies, short stature, and talipes equinovarus at birth (Figure 2). Apart from the facial muscle weakness, these features were absent in our patient. An operation under general anesthesia has never been performed on the proband. In addition, the patient does not have a history of malignant hyperthermia, which is typical for patients with NAM [3,5,6]. A large amount of patients with NAM suffer from ventilatory problems and respiratory distress in infancy. Moreover, in this case, some patients receive a non-invasive ventilator support [1,5]. Our patient was hospitalized with pneumonia due to aspiration at the age of 3 days and required mechanical ventilation until the age of 4.5 months. It is highly possible that this episode is associated with the major disorder.

An additional prominent manifestation of NAM is severe early-onset of progressive scoliosis, which often requires surgical correction [3,5]. Our patient has severe spine problems, which are partially corrected by conservative methods without surgery. Due to the severe scoliosis, our patient developed a tilted pelvis that was reflected in the peculiar gait of the girl. In addition, the proband had a retraction of the ankle tendons that was surgically corrected. More than half of the infants with NAM have talipes equinovarus at birth (Figure 2). Our patient did not have any contractures at birth. Nevertheless, on clinical examination, pes cavus of both sides was noted, which probably developed due to the ankle tendon retractions (Figure 1).

Our patient was initially suspected of having some form of congenital myopathy due to the absence of a certain specific phenotype of NAM. Therefore, she was referred to the sequencing of genes associated with congenital myopathies. Two compound heterozygous SNVs in the *STAC3* gene were revealed. One of the SNVs c.862A>T, p.(Lys288Ter) was previously reported in a patient of Turkish ancestry [4]. Another novel variant c.93del leads to a frameshift with a premature stop codon at position 78 (p.Lys32ArgfsTer78). Both SNVs are pathogenic according to the ACMG criteria [7].

The *STAC3* gene encodes SH3 and cysteine-rich domain 3, a component of excitation-contraction coupling (ECC) in skeletal muscle [1]. It is expressed almost exclusively in skeletal muscle. Along with other proteins, such as Cav1.1, RyR1, Cavβ1a, and junctophilin, STAC3 is responsible for the proper working of the skeletal muscle ECC machinery [8]. Nelson et al. showed that STAC3-null mice, which are completely paralyzed, die at birth due to asphyxia [9].

It is shown that STAC3 directly interacts with Cav1.1. In addition, the amino acid substitution of the STAC3 protein Trp284Ser, Phe295Leu, and Lys329Asn affects the ECC by violating the interaction with the II–III loop of Cav1.1 [10]. All of these mutant isoforms reduce the affinity to Cav1.1 and disrupt the ECC calcium transients, while the isoform with the Trp284Ser substitution has the most negative effect on binding with Cav1.1 [8] and is associated with a more severe phenotype.

It seems interesting that the phenotype of our proband is similar to the phenotype of the patient of Turkish ancestry with the same nonsense variant and splice donor site variant c.432+4A>T [4]. The authors showed that c.432+4A>T leads to two alternative splicing events, one causes in-frame deletion due to skipping of exon 4, and another due to the activation of a new cryptic splice donor site, leads to insertion of the part of intron 4 with frameshift.

To our knowledge, we reported the first case of a patient who has both nonsense and frameshift SNVs. It is assumed that the frameshift variant and premature stop codon that lead to the nonsense-mediated RNA decay was followed by an absence of the protein product, which is supposed to result in a more severe clinical picture. However, our patient has a milder phenotype compared to the reported patients with frequent homozygous loss-of-function missense variant Trp284Ser. To the best of our knowledge, the *STAC3* gene has seven isoforms, which are all expressed in muscle tissue at different levels, according to the GTEx data, and four are coding proteins (Figure 3). The most expressed isoform is ENST00000332782.7 (STAC3-201). The second isoform that is expressed at a high level is ENST00000546246.2 (STAC3-202). The remaining isoforms are expressed several times less than the previous ones. We assume that isoform STAC3-202 is not affected by the frameshift variant as the isoform STAC3-206 (Figure 3). The frequent variant p.Trp284Ser affects these three protein-coding isoforms. The shortest isoform STAC3-204 could still be functional from the allele with the variant p.Lys288Ter in our patient. However, this isoform is also not affected by the frequent missense variant p.Trp284Ser. In this way, we can speculate that partially, functional isoforms STAC3-202 and STAC3-206, could explain the milder phenotype of our patient. Possibly, the same explanation is applicable for the Turkish case with intronic SNV that does not affect isoform STAC3-202. However, the activation of a new cryptic splice donor site could lead to different alterations in all of the isoforms. Undoubtedly, this hypothesis needs further functional investigation.

Similar to the other non-Amerindian NAM cases previously described, our report shows that NAM should be considered in patients with congenital myopathy of any ancestry. Even though it is possible to mark the peculiar features of this myopathy, there may be a milder phenotype, which is difficult to distinguish from other congenital myopathies.

## Figures and Tables

**Figure 1 genes-13-00341-f001:**
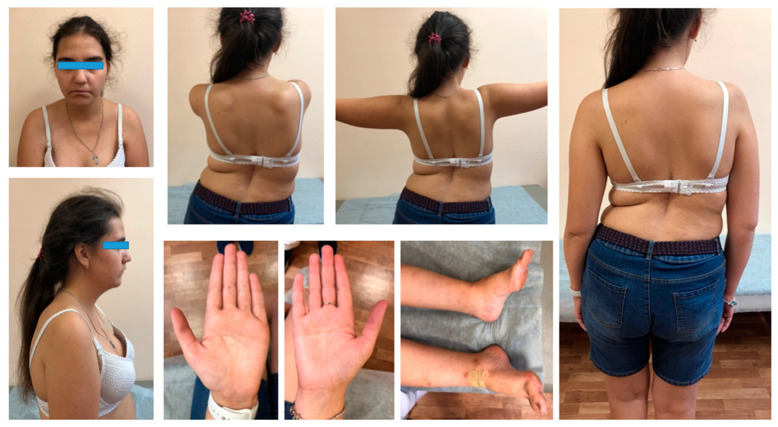
Patient at 17 years demonstrating downturned corners of the mouth, severe scoliosis, atrophy of pectoralis muscles and upper limb girdle muscles, mild clinodactyly of the right fifth finger, pes cavus, and asymmetric hip position.

**Figure 2 genes-13-00341-f002:**
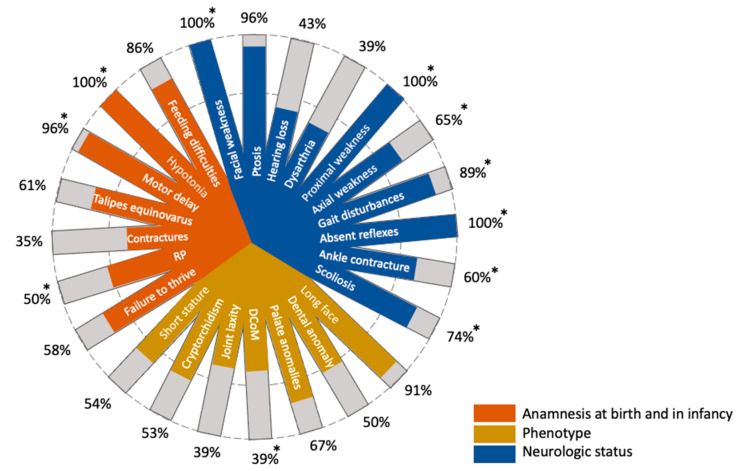
Clinical features of the patients with genetically confirmed NAM, as presented in the literature [3,4,5]. Asterisks indicate the features detected in our patient.

**Figure 3 genes-13-00341-f003:**
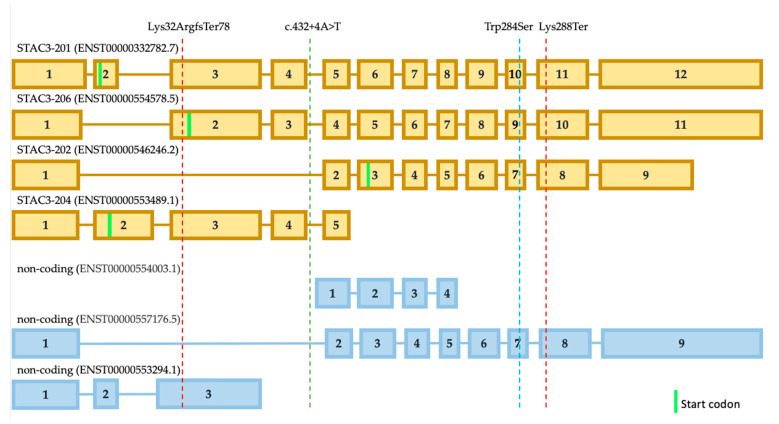
The predicted localization of SNVs on coding and non-coding isoforms of the *STAC3* gene. The red lines indicate the SNVs revealed in the presented case, the green line indicates the SNVs detected in the patient with a Turkish ancestry [4], and the blue line corresponds to a frequent pathogenic variant.

**Table 1 genes-13-00341-t001:** Clinical features of non-Amerindian patients with *STAC3* variants.

	Grzybowski et al. 2017 [4]	Telegrafi et al. 2017 [3]				Current Publication
Genotype (ENST00000332782.7)	c.862A>T (p.Lys288Ter); c.432+4A>T	c.851G>C (p.Trp284Ser); c.851G>C (p.Trp284Ser)		c.851G>C (p.Trp284Ser); 763_766delCTCT (p.Leu255IlefsTer58)		c.862A>T (p.Lys288Ter); c.93del (p.Lys32ArgfsTer78)
Ancestry	Turkish	Qatari		Puerto Rican		Russian
Age (Years)/gender	19/M	8/F	18/F	21/M	16/F	17/F
Hypotonia at birth	+	+	+	+	+	+
Motor delay	+	+	+	+	+	+
Feeding difficulties	+	+	n/d	+	+	−
Long face	−	+	n/d	+	+	+
Facial weakness	+	+	+	+	+	+
Ptosis	+	+	+	+	+	−
Limited extraocular movements	−	−	+	+	−	−
Micrognathia	+	−	n/d	−	+	+
Palate anomalies	+	+	+	+	+	−
Mouth usually in open position	−	+	n/d	−	+	+
Downturned corners of the mouth	+	+	n/d	+	+	+
Low-set ears	+	+	n/d	+	−	−
Hearing loss	−	+	+	+	−	−
Dysarthria	−	n/d	+	+	+	−
Decreased skin creases	−	+	n/d	+	+	−
Short stature	+	+	+	+	+	−
Scoliosis	+	+	+	+	+	+
Pectus excavatum	−	+	n/d	−	−	+
Contractures	ankle	n/d	n/d	n/d	n/d	ankle
Cavus foot	−	n/d	n/d	−	−	+
Overlapping toes	−	n/d	n/d	+	+	−
Axial weakness	−	+	n/d	+	+	+
Proximal weakness	+	+	+	+	+	+
Weakness distal > proximal	−	−	n/d	+	−	−
Gowers’ sign	+	n/d	n/d	−	+	+
Gait disturbances	+	n/d	+	−	+	+
Ventilatory failure episodes	+	−	−	+	+	1 (in the first month)
History of MH	−	−	+	+	−	−
CK level	normal	n/d	n/d	normal	n/d	normal

M: Male; F: Female; n/d: No data; MH: Malignant hyperthermia; CK: Creatine kinase.

## Data Availability

No additional data sets are associated with this paper.

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
