# Peer review of "The First Russian Patient with Native American Myopathy"

_genes, 2022, doi:10.3390/genes13020341_

Round 1

Reviewer 1 Report

The authors report a Russian patient with STAC3 mutated myopathy due to compound heterozygous truncating variants. The paper is of interest because of the rarity of the disease outside the Native American community and some new features including milder presentation compared to the original syndromic patients.

Concerns:
- the medical English language needs some polishing
- a muscle MRI study would be an excellent complement to the clinical study results as such findings would add important novelty to the paper

Author Response

Point 1: The medical English language needs some polishing.

Response 1: Thank you for your precise remarks. We have corrected both spelling errors and inaccuracies in medical English.

Point 2: A muscle MRI study would be an excellent complement to the clinical study results as such findings would add important novelty to the paper.

Response 2: Unfortunately, the proband lives in a small town where muscle MRI is not performed, and she is not planning to visit our facility in the nearest future due to an epidemic situation.

Reviewer 2 Report

The authors provide a case report with a thorough description of a young woman of Russian origin. A genetic diagnosis of NAM is suggested based on segregation of the variants in the relatively small family (parents and healthy brother). One of the two identified mutations is reported earlier, whereas the other is novel.

In general I find the manuscript to be a nice case report, but have two minor comments I'd like addressed:

  • I would like to see a little more discussion on the isoform(s) that are not affected by the novel c.93del variant. Is anything known about their function? The Turkish splice mutation also does not affect all isoforms. How does this compare to the c.93del variant in this report?
  • The manuscript should be fixed for grammar and readability. Gene names should also be in italics.

Author Response

Point 1: I would like to see a little more discussion on the isoform(s) that are not affected by the novel c.93del variant. Is anything known about their function? The Turkish splice mutation also does not affect all isoforms. How does this compare to the c.93del variant in this report?

Response 1: Thank you for your important notes. We have added more discussion on the isoforms as well as a new figure reflecting the coding and non-coding isoforms for a more precise explanation of our hypothesis. Unfortunately, we did not find any publications on the functions of other isoforms. However, our assumption remains valid. The STAC3 gene has 7 isoforms, all of them expressed in muscle tissue at different levels according to GTEx data. One of the coding isoforms expressed at a high level is ENST00000546246.2 (STAC3-202), and we assume that it is not affected by frameshift variant like another coding isoform STAC3-206. The frequent variant p.Trp284Ser affects the main isoform and these two protein-coding isoforms. The shortest isoform STAC3-204 could be still functional from allele with the variant p.Lys288Ter in our patient, but this isoform is also not affected by frequent missense variant p.Trp284Ser. In this way, we can speculate that partially functional isoforms STAC3-202 and STAC3-206 could explain the milder phenotype of our patient. The same explanation is probably applicable for the Turkish case with intronic SNV that does not affect isoform STAC3-202. However, the activation of a new cryptic splice donor site could lead to different alterations in all isoforms. Undoubtedly, this hypothesis needs further functional investigation

Point 2: The manuscript should be fixed for grammar and readability. Gene names should also be in italics.

Response 2: We have corrected the text for grammar and readability. Unfortunately, we have missed that gene names were not in italics. That was a technical issue because of the transfer of the text from one file to another.